# Fruit Juice Consumption, Body Mass Index, and Adolescent Diet Quality in a Biracial Cohort

Lynn L. Moore [1,*], Xinyi Zhou [1], Li Wan [1,†], Martha R. Singer [1], M. Loring Bradlee [1] and Stephen R. Daniels [2]

1 Department of Medicine/Preventive Medicine and Epidemiology, Boston University Chobanian and Avedisian School of Medicine, Boston, MA 02118, USA

2 Department of Pediatrics, University of Colorado School of Medicine, Aurora, CO 80045, USA

* Correspondence: llmoore@bu.edu; Tel.: +1-617-358-1325

† Current address: Data Sciences Program, University of California-Los Angeles, Los Angeles, CA 90095, USA.

**Abstract:** Fruit juice consumption during childhood remains controversial. Here, we evaluated the association between preadolescent 100% fruit juice intake and later adolescent diet quality and body mass index (BMI). We used prospective data over 10 years from the National Growth and Health Study for 1921 black and white girls, ages 9–10 years at baseline, for analyses of diet quality, and 2165 girls for BMI analyses. Statistical analyses included repeated measures analysis of variance and logistic regression models. Girls who drank $\geq 1.0$ cup/day of fruit juice in preadolescence consumed 0.44 cup/day more total fruit in later adolescence than non-juice-drinking girls ($p < 0.0001$). White and black girls who drank $\geq 1.25$ cups/day in preadolescence were 2.62 (95% CI: 1.35–5.08) and 2.54 (1.27–5.07) times more likely, respectively, to meet the Dietary Guidelines for whole fruit by later adolescence than those with the lowest juice intakes. Further, fruit juice consumption was positively associated with diet quality scores. Overall, girls consuming $\geq 1.25$ cups/day of juice had a BMI in late adolescence that was 1.7 kg/m$^2$ lower than that of non-juice-drinking girls. In conclusion, early adolescent fruit juice intake was positively associated with subsequent whole fruit consumption, better diet quality, and lower BMI in later adolescence.

**Keywords:** fruit juice; fruit intake; diet quality; pediatric obesity; BMI; adolescence; Healthy Eating Index

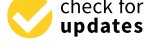



## 1. Introduction

Fruit is an important source of beneficial nutrients and has anti-inflammatory and antioxidant properties [1]. Fruit consumption has also been linked with positive health outcomes, including reduction in risk of hypertension and cardiovascular diseases [2,3]. Current Dietary Guidelines for Americans (DGA) recommend that at least half of the recommended total daily fruit intake for both children and adults should be derived from whole fruit [4]. While the nutrient composition of whole fruit and fruit juice is very similar, whole fruit is considered by some to be superior to 100% fruit juice due to its higher fiber content and slower absorption in the gut [5]. Data from the National Health and Nutrition Examination Survey (NHANES) suggest that children and adolescents generally fail to meet the DGA recommendations for total fruit consumption [6,7]. Although one analysis suggests that whole fruit intake has improved over time [8], the declining intakes of fruit juice over the last few decades may be responsible for the failure of most children to meet the current recommendations for total fruit consumption [9]. It is possible that fruit juice consumption may encourage the later consumption of whole fruit, although data on its contribution to whole fruit intake are limited. Recent data from NHANES found that children who consumed both more milk and more 100% fruit juice had higher scores on the Healthy Eating Index (HEI) [10].

Fruit juice consumption among children is controversial primarily due to concerns about its energy content and the resulting potential for excess weight gain and metabolic

disturbances [11]. At least one recent study found fruit juice consumption to be associated with higher levels of abdominal adiposity [12]. However, a 2016 review of 22 studies found no independent association between 100% fruit juice and risk of childhood obesity [13], and another more recent review also concluded that 100% fruit juice was not generally associated with excess weight gain or other metabolic problems during childhood [14]. Recent data from the Growing Up Today Study (GUTS) found an inverse association between orange juice consumption and body mass index (BMI) in girls and a null association in boys [15]. Another analysis in that same study also found orange juice consumption to be positively associated with height gain but not with weight gain [16].

A 2019 study of 100% fruit juice intake and diet quality among adults in the NHANES cohort concluded that fruit juice consumption was positively associated with diet quality [17]. A 2017 review of this topic by an expert panel concluded that children consuming higher amounts of 100% fruit juice had better diet quality, with higher intakes of folate, vitamin C, and potassium [18]. In another such study, children who consumed more 100% fruit juice were more likely to have adequate intakes of dietary fiber, potassium, magnesium, and vitamin C [14].

In this study, data from the prospective National Heart, Lung, and Blood Institute's National Growth and Health Study (NGHS) were used to evaluate the association between 100% fruit juice consumption in preadolescence and subsequent intakes of whole and total fruit in white and black girls throughout adolescence. We hypothesized that girls who consumed more fruit juice at baseline would have higher scores on the Healthy Eating Index (HEI) [19] and a higher likelihood of meeting the DGA for fruit consumption. In addition, we theorized that girls consuming more fruit juice would have no greater gains in BMI during adolescence than those who consumed less.

## 2. Materials and Methods

### 2.1. Subjects

The NGHS is a prospective study that began in 1987 with the recruitment and enrollment of 2379 9–10 year old (preadolescent) girls. There were approximately equal numbers of black and white girls, and they were followed annually for 10 years. In an attempt to provide a representative sample of urban residential and suburban families, participants were recruited from three clinical centers: the University of California at Berkeley, Berkeley, CA, USA; the University of Cincinnati–Cincinnati Children's Hospital Medical Center, Cincinnati, OH, USA; and Westat, Inc., Rockville, MD, USA (which was associated with a Washington, DC metropolitan-area health maintenance organization). Girls who self-declared as black or white and whose parents were similarly self-identified were eligible to participate if they were 9–10 years of age at the time of the first clinic visit and their parents or guardians signed an informed consent. Details of the original study design and methods have been previously published [20]. Of the 2379 girls enrolled at baseline, 36 failed to provide dietary data at the first visit and an additional 422 girls were missing dietary data through the end of follow-up (at age 17 or older), leaving 1921 girls for the analysis of fruit juice consumption and diet quality. For the analyses of juice intake and BMI, we excluded 36 girls missing dietary data at baseline and 178 girls with missing BMI at age 17 or older. This left 2165 girls for the BMI analyses.

The original NGHS protocol was approved by the Institutional Review Board (IRB) of each participating clinical center. These secondary analyses were deemed exempt by the Boston University Medical Campus IRB. The original data used in this study are publicly available through the NHLBI's BioLINCC repository (Biologic Specimen and Data Repository Information Coordinating Center (BioLINCC, Bethesda, MD, USA), RRID:SCR_013142).

### 2.2. Dietary Intake

Each child's diet was evaluated using 3-day diet records during 8 of the 10 years of follow-up (years 1–5, 7, 8, and 10). Participants received instructions for completing the

diet records from a trained study nutritionist using age-appropriate language. Girls were taught to record all food and drink consumed on three consecutive days, including two weekdays and one weekend day. The girls used standard measuring cups, spoons, and geometric shapes to estimate portion sizes and, whenever necessary, received information from a parent on recipes, brands, and other details of the foods consumed. A standardized debriefing was carried out for each set of dietary records. Data were entered into the Nutrition Data System (NDS) of the University of Minnesota, Minneapolis, MN (Food Table Version 19 was used at baseline, with updated versions of the database being used as the study progressed) [21]. During the first two years, data were entered at the Nutrition Coordinating Center (NCC, Minneapolis, MN, USA) of the University of Minnesota while in subsequent years, data were entered onsite by a NCC-trained nutritionist. Data on food servings including cup-equivalents of whole (raw) fruit (e.g., apples, pears, citrus fruits, melons, and berries), 100% fruit juice (consumed as a beverage), and total fruit (from all sources) were derived by the authors (M.R.S. and L.L.M.) by linking NDS food codes with USDA food codes [22]. Sweetened juices, such as cranberry juice and part-juice drinks, were excluded from the category of 100% fruit juice in these analyses, but the fruit portion of a part-juice drink (e.g., the fruit in a fruit smoothie) was counted in the total fruit category. Intakes of total fruit, whole fruit, and 100% fruit juice were estimated as the mean intake across all available days of dietary data within each of the following age groups: 9–10, 11–12, 13–14, 15–16, and 17–20 years. There were 186 girls who provided the first set of dietary records after they had turned 11 years old because the dietary records were completed between the baseline visit (when the girls were 9–10 years of age) and the second visit (when they were 11–12 years of age). Data for these girls were included in the calculation of baseline fruit juice intake.

*2.3. Diet Quality*

The HEI is a measure of diet quality that was designed through the collaborative efforts of the USDA (Washington, DC, USA) and the National Cancer Institute (Bethesda, MD, USA) to evaluate the extent to which an individual's dietary intake met the recommendations of the DGA. The HEI is comprised of 13 component scores with a maximum total score of 100. Fruit is included in two components: total fruit and whole fruit, with total fruit including both whole fruit and 100% fruit juice. HEI scores were calculated using the HEI Scoring Algorithm available at https://epi.grants.cancer.gov/hei/sas-code.html. (accessed on 4 May 2023)

*2.4. Body Mass Index (BMI)*

Height was measured annually in duplicate using a portable stadiometer, and weight was measured on a digital scale. BMI was estimated as mean weight (in kilograms) at each exam divided by mean height (in squared meters).

*2.5. Statistical Analysis*

Each child's mean baseline intake of 100% fruit juice was calculated using all available days of diet records during the first examination visit year. Intake was classified into one of four categories of intake for most analyses: 0 cups, >0–<0.5 cup/day, 0.5–<1.0 cup/day, and ≥1.0 cup/day. Because the recommended juice intake for girls of this age is up to 0.75 cup/day, we first chose our categories to bracket this level of intake and to compare it with higher and lower intakes. The primary contrast to be examined was the highest (≥1.0 cup/day) vs. the lowest (0 cups/day) juice intake. Sensitivity analyses were used to explore the sensitivity of the results to changes in juice intake categories. A second set of cutoff values (0 cups, >0–<0.75 cup/day, 0.75–<1.25 cup/day, and ≥1.25 cups/day) was designed to determine whether even higher intakes of juice (≥1.25 cups/day) had any adverse effects, particularly on BMI.

To address the first analytic question of the impact of fruit juice intake in preadolescence on the consumption of total fruit and whole fruit throughout adolescence, we used

mixed models for repeated measures due to the unbalanced nature of the groups. Mixed models allow for inclusion of both fixed and random effects and incorporate an interaction term for age (group) by juice intake (group). In these analyses, data were missing at random over the course of follow-up. Further, we chose to use an unstructured covariance assumption. Similar models were used to evaluate the association between early juice consumption and total HEI scores as well as mean BMI throughout adolescence. Logistic regression models were used to estimate the likelihood of meeting the current DGA for whole and total fruit intakes in later adolescence in each of the fruit juice intake categories, with a primary focus on comparing the highest vs. the lowest intakes. Because several of these outcomes, including intakes of whole fruit and fruit juice as well as BMI, have been shown to differ by race, we also carried out these analyses separately for black and white girls [23].

We explored potential confounding by several factors including baseline age, race (for combined models), socioeconomic status (SES) based on parental education level, physical activity, baseline BMI, total energy intake, and percent of calories from carbohydrates, protein, and fats. We examined each factor alone as a potential confounder and then in combination with other factors. We were careful not to include factors that were likely to be collinear in the same model (e.g., including carbohydrates, protein, and dietary fats in the same model). We found no confounding by any of these factors, alone or in combination, as indicated by an observed change of generally less than approximately 1% in the effect estimates. Nonetheless, we included race in the combined models but focused the primary results on the race-specific models.

## 3. Results

In Table 1, participant characteristics are given in four categories of intake of 100% fruit juice at baseline. There was no statistically significant difference in percent body fat or BMI at baseline by category of 100% fruit juice intake. Girls consuming more fruit juice tended to have lower energy-adjusted intakes of protein and fat and higher intakes of carbohydrates. Those girls who drank more fruit juice in preadolescence also tended to consume more whole fruit at that age than those drinking less.

**Table 1.** Baseline descriptive characteristics of girls according to 100% fruit juice intake at baseline.

| | Intake of 100% Fruit Juice at Baseline | | | | |
| | 0 Cups *n* = 535 | >0–<0.5 Cup *n* = 745 | 0.5–<1.0 Cup *n* = 431 | ≥1.0 Cup *n* = 210 | |
|---|---|---|---|---|---|
| | Mean ± s.d. | | | | *p*-trend |
| Age (years) | 10.5 ± 0.57 | 10.3 ± 0.47 | 10.3 ± 0.48 | 10.3 ± 0.50 | <0.0001 |
| Height (meters) | 1.43 ± 0.07 | 1.42 ± 0.07 | 1.43 ± 0.07 | 1.44 ± 0.07 | 0.0275 |
| BMI (kg/m$^2$) | 19.0 ± 4.0 | 18.9 ± 3.7 | 18.4 ± 3.7 | 18.7 ± 3.7 | 0.1396 |
| Body fat (%) | 24.9 ± 7.2 | 24.6 ± 7.3 | 23.9 ± 6.7 | 23.9 ± 7.0 | 0.1163 |
| Energy (kcals/day) | 1791 ± 463 | 1809 ± 454 | 1903 ± 444 | 2033 ± 459 | <0.0001 |
| Protein (% of energy) | 14.6 ± 2.8 | 14.3 ± 2.5 | 13.9 ± 2.5 | 13.8 ± 2.3 | <0.0001 |
| Total fat (% of energy) | 36.5 ± 5.3 | 36.3 ± 4.8 | 35.0 ± 4.9 | 33.5 ± 4.5 | <0.0001 |
| Carbohydrates (% of energy) | 50.0 ± 6.7 | 50.5 ± 5.9 | 52.2 ± 6.1 | 54.0 ± 5.4 | <0.0001 |
| Calcium (mg/day) | 806 ± 297 | 798 ± 284 | 799 ± 281 | 844 ± 310 | 0.2547 |
| Potassium (mg/day) | 1942 ± 592 | 1964 ± 553 | 2146 ± 537 | 2513 ± 590 | <0.0001 |
| Whole fruit (cup-equivalents/day) | 0.45 ± 0.60 | 0.49 ± 0.55 | 0.53 ± 0.56 | 0.65 ± 0.74 | 0.0009 |
| 100% fruit juice (cup-equivalents/day) | 0.0 ± 0.0 | 0.26 ± 0.11 | 0.67 ± 0.14 | 1.37 ± 0.48 | <0.0001 |
| Race (*n*, % white) | 263 (49%) | 363 (49%) | 215 (50%) | 95 (45%) | 0.9214 |
| Socioeconomic status (*n*, % low) | 116 (22%) | 187 (25%) | 93 (22%) | 34 (16%) | 0.2357 |

Figure 1 provides descriptive information on the median intakes of total fruit as well as whole fruit and 100% fruit juice for white and black girls throughout adolescence. The highest median intake of fruit, whether juice or whole fruit, was found at 9–10 years of age. Black girls consumed less total fruit than white girls at all ages, and this finding was attributable to lower intakes of whole fruit. Supplementary Table S1 also shows the median intake values and interquartile ranges for the intakes at each age in Figure 1.

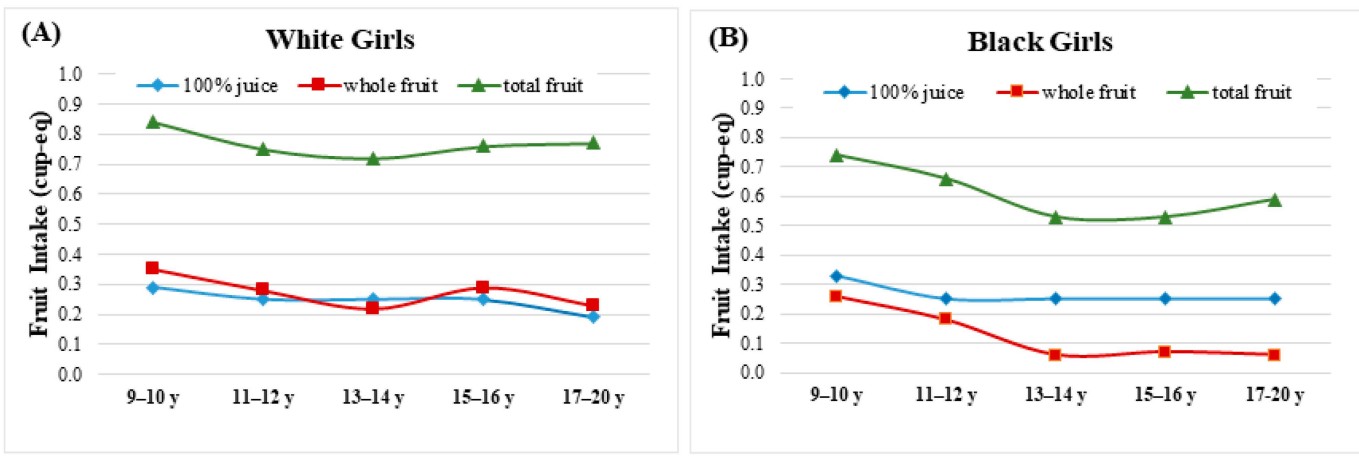

**Figure 1.** Median intakes of total fruit (green lines with triangles), whole fruit (red lines with squares), and 100% fruit juice (blue lines with diamonds) consumption throughout adolescence, in white (Panel (**A**)) and Black (Panel (**B**)) girls.

Figure 2 shows the association between the categories of 100% fruit juice consumption at baseline and intakes of total fruit (Panel A), whole fruit (Panel B), and 100% fruit juice (Panel C) from preadolescence to the end of adolescence. The mean intakes of total fruit throughout adolescence (Panel A) were highest among those girls with higher levels of juice consumption at baseline. Specifically, preadolescent girls who drank $\geq$ 1.0 cup/day of 100% fruit juice consumed 0.44 more cup/day of total fruit at 17–20 years of age than those who did not drink fruit juice at baseline (mean total fruit intakes at ages 17–20: 1.19 vs. 0.75 cup/day in the highest vs. lowest juice-drinking categories, respectively), as shown in Supplementary Table S2. In addition, girls who consumed the most fruit juice at baseline had the highest intakes of whole fruit at the end of adolescence (Panel B). Supplementary Table S2 also shows that overall, girls who drank more fruit juice at baseline had statistically significantly higher intakes of total fruit at every follow-up age period.

We also examined the association between 100% fruit juice consumption at baseline and subsequent overall diet quality scores as measured by the HEI in Figure 3. The sample sizes in each juice-drinking category at each age are shown below the x-axis labels. The figure shows that among white girls, the total HEI scores at each age were positively associated with fruit juice intake categories ($p < 0.0001$ at all ages). These associations among black girls were similar although somewhat less consistent during mid-adolescence. For white girls, the HEI scores at the end of follow-up (ages 17–20 years) increased as the amount of fruit juice consumed at baseline also increased. For black girls, the HEI scores at the end of follow-up were higher among girls who drank at least 0.5 cup/day at baseline, with those who drank >1.0 cup/day having the highest HEI scores ($p = 0.054$, comparing girls drinking >1.0 cup/day with those who did not drink juice at baseline).

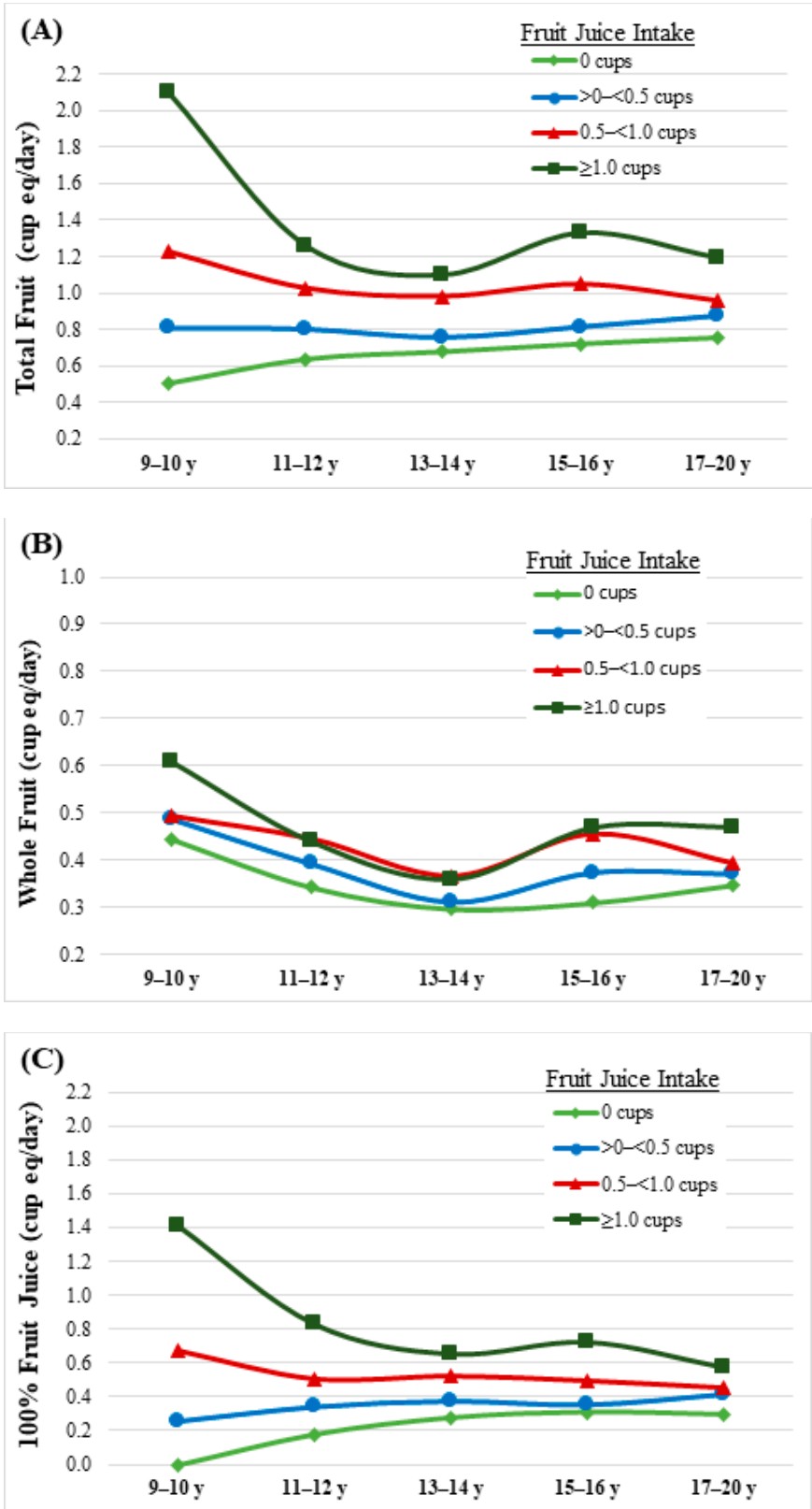

**Figure 2.** Mean intakes, adjusted for race, for total fruit (Panel (**A**)), whole fruit (Panel (**B**)), and 100% fruit juice (Panel (**C**)) at each age during adolescence in four categories of 100% fruit juice intake at preadolescence.

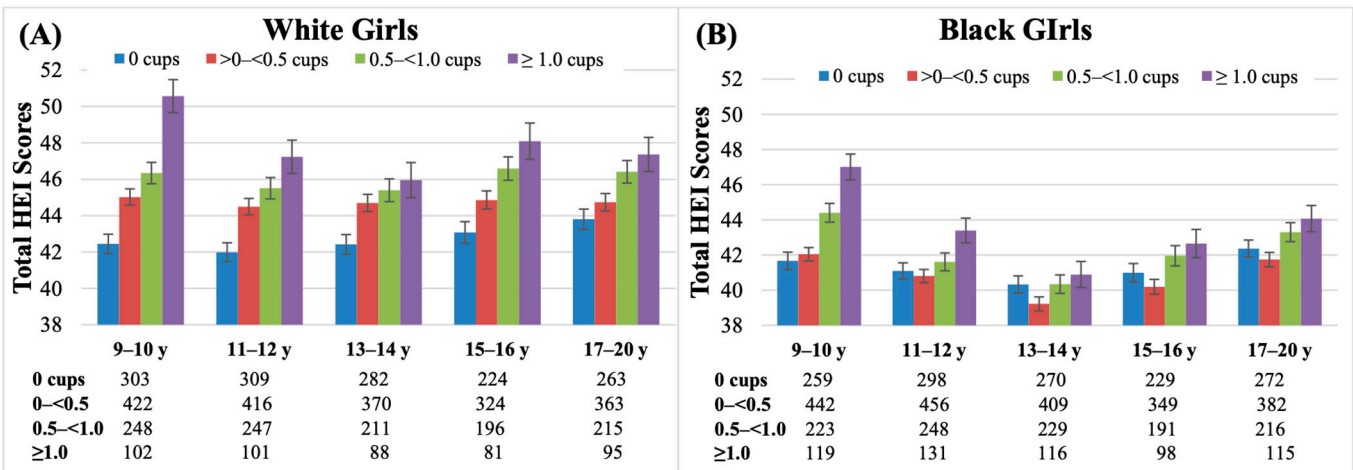

| Fruit Juice Intake (Baseline) | White girls | | Black girls | |
|---|---|---|---|---|
| | Overall Differences (p-values) | End of Follow-up (p-values) | Overall differences (p-values) | End of Follow-up (p-values) |
| 0 versus <0.5 cups | <0.0001 | 0.202 | 0.179 | 0.322 |
| 0 versus 0.5–<1.0 cups | <0.0001 | 0.002 | 0.014 | 0.199 |
| 0 versus ≥1 cup | <0.0001 | 0.001 | <0.0001 | 0.054 |

**Figure 3.** Healthy Eating Index scores throughout adolescence according to 100% fruit juice intake categories in preadolescence among white (Panel (**A**)) and black (Panel (**B**)) girls. Sample sizes for each age period are shown below the x-axis age groups and show the number of girls providing dietary data for the calculation of HEI scores in each age and juice-drinking category. Note that some girls completed their first food diaries at age 11, leading to slightly larger sample sizes in the 11–12-year-old age group.

Table 2 evaluates the likelihood of meeting the DGA [24] at the end of adolescence for total fruit (≥1.5 cups/day) and whole fruit (≥0.75 cup/day) intakes according to 100% fruit juice consumption at baseline. We show two different exposure categories for the highest juice intake in this table: ≥1.0 cup/day and ≥1.25 cups/day. Since only 10 white girls and 13 black girls drank more than 2.0 cups/day of 100% fruit juice, we were unable to evaluate intakes at that level as a separate category. We found that girls who drank ≥1.0 cup/day of fruit juice (vs. non-juice consumers) in preadolescence were 2.48 times as likely (95% CI: 1.70–3.61) to meet dietary recommendations for total fruit and 2.12 times as likely (95% CI: 1.44–3.12) to meet dietary recommendations for whole fruit at 17–20 years of age. Girls who drank ≥1.25 cups/day of juice were even more likely to meet the DGA for whole fruit at older ages. There were few race-specific differences in these results.

Figure 4 and Table 3 examine the relation between fruit juice consumption and adolescent BMI. Trends in BMI according to fruit juice intake in preadolescence are shown first in Figure 4. Here, we see that girls with the highest baseline juice intake had a lower BMI throughout adolescence than girls with the lowest baseline juice intake (*p* = 0.0063 and *p* = 0.0143 comparing highest vs. lowest for white and black girls, respectively). At the end of adolescence (Table 3), girls who consumed ≥1.25 cups/day of juice had lower BMI levels (2.2 kg/m$^2$ lower for white girls and 1.5 kg/m$^2$ lower in black girls) than non-juice drinkers. In fact, the highest BMIs at the end of adolescence in both white and black girls were found among nonfruit juice consumers, and the lowest BMIs were found in the highest juice consumers. In Supplementary Table S3, we further examined whole fruit intake as well as total fruit intake at baseline in association with BMI at the end of follow-up. In these analyses, we found that the association between whole fruit and BMI was slightly weaker than that observed in Table 3 for 100% fruit juice and subsequent BMI.

**Table 2.** Race-specific likelihood of meeting dietary guidelines for total fruit and whole fruit in late adolescence according to categories of fruit juice intake at baseline.

| Fruit Juice Intake | N | Number (%) Meeting Guidelines | All Subjects | | White | | Black | |
|---|---|---|---|---|---|---|---|---|
| | | | OR | 95% CI | OR | 95% CI | OR | 95% CI |
| Intake Categories | | | **Meeting the DGA for Total Fruit** | | | | | |
| 0 cups | 535 | 82 (15.3%) | 1.00 | (Ref) | 1.00 | (Ref) | 1.00 | (Ref) |
| <0.5 cup | 745 | 137 (18.4%) | 1.25 | 0.92–1.68 | 1.58 | 1.06–2.35 | 0.9 | 0.56–1.43 |
| 0.5–<1.0 cup | 431 | 94 (21.8%) | 1.54 | 1.11–2.14 | 1.91 | 1.24–2.95 | 1.14 | 0.68–1.91 |
| ≥1.0 cups | 210 | 65 (31.0%) | 2.48 | 1.70–3.61 | 2.88 | 1.71–4.85 | 2.21 | 1.28–3.82 |
| Intake Categories | | | | | | | | |
| 0 cups | 535 | 82 (15.3%) | 1.00 | (Ref) | 1.00 | (Ref) | 1.00 | (Ref) |
| >0–<0.75 cup | 1048 | 199 (19.0%) | 1.30 | 0.98–1.72 | 1.61 | 3.37–2.34 | 0.97 | 0.63–1.49 |
| 0.75–<1.25 cups | 229 | 59 (25.8%) | 1.92 | 1.31–2.80 | 2.45 | 1.49–4.05 | 1.38 | 0.76–2.51 |
| ≥1.25 cups | 109 | 38 (34.9%) | 2.96 | 1.87–4.68 | 3.37 | 1.75–6.49 | 2.74 | 1.43–5.27 |
| Intake Categories | | | **Meeting the DGA for Whole Fruit** | | | | | |
| 0 cups | 535 | 80 (15.0%) | 1.00 | (Ref) | 1.00 | (Ref) | 1.00 | (Ref) |
| <0.5 cup | 745 | 120 (16.1%) | 1.09 | 0.80–1.49 | 1.27 | 0.86–1.89 | 0.86 | 0.52–1.42 |
| 0.5–<1.0 cup | 431 | 82 (19.0%) | 1.34 | 0.95–1.87 | 1.47 | 0.95–2.27 | 1.16 | 0.67–2.00 |
| ≥1.0 cup | 210 | 57 (27.1%) | 2.12 | 1.44–3.12 | 2.55 | 1.52–4.28 | 1.84 | 1.01–3.34 |
| Intake Categories | | | | | | | | |
| 0 cups | 535 | 80 (15.0%) | 1.00 | (Ref) | 1.00 | (Ref) | 1.00 | (Ref) |
| >0–<0.75 cup | 1048 | 171 (16.3%) | 1.11 | 0.83–1.48 | 1.23 | 0.84–1.78 | 0.96 | 0.61–1.52 |
| 0.75–<1.25 cups | 229 | 55 (24.0%) | 1.80 | 1.22–2.64 | 2.45 | 1.50–4.01 | 1.08 | 0.55–2.11 |
| ≥1.25 cups | 109 | 33 (30.3%) | 2.47 | 1.54–3.96 | 2.62 | 1.35–5.08 | 2.54 | 1.27–5.07 |

DGA: Dietary Guidelines for Americans; OR: Odds Ratio; CI: Confidence Interval; Ref: Reference group; DGA: Dietary Guidelines for Americans.

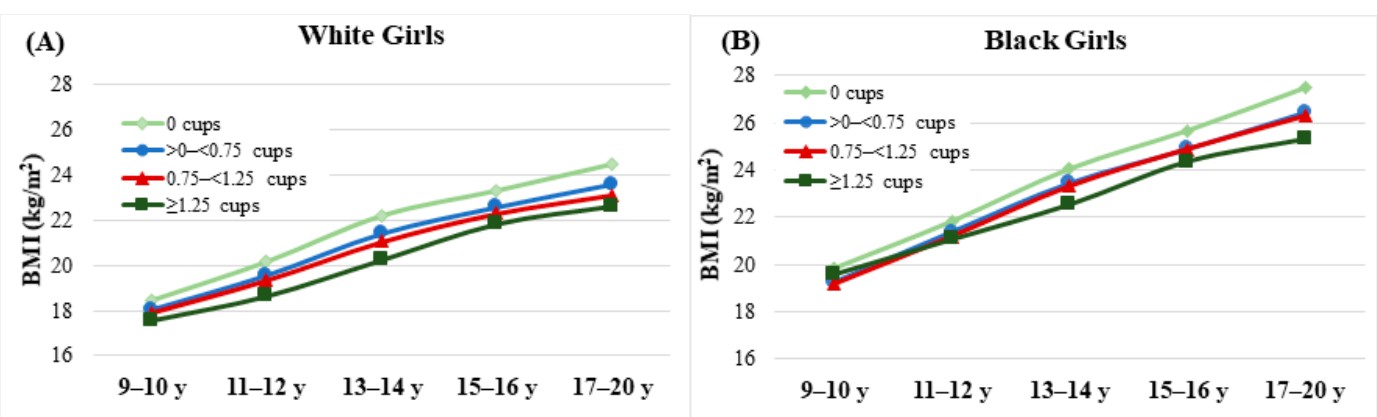

**Figure 4.** Trends in BMI throughout adolescence according to fruit juice intake at baseline for white (Panel (**A**)) and black (Panel (**B**)) girls. *p*-values comparing the change in BMI associated with the lowest vs. highest juice intakes were 0.0063 and 0.0143 for white and black girls, respectively.

**Table 3.** Mean BMI at 17–20 years of age according to intake of 100% fruit juice at baseline.

| Baseline Juice Intake | *n* | | All Girls * *n* = 2165 | White Girls *n* = 1052 | Black Girls *n* = 1113 |
|---|---|---|---|---|---|
| | | Median | | Mean ± s.e. | |
| 0 cups | 601 | 0 cups | 25.8 ± 0.26 | 24.4 ± 0.29 | 27.2 ± 0.42 |
| >0–<.75 cup | 1187 | 0.33 cup | 25.1 ± 0.19 | 23.6 ± 0.21 | 26.4 ± 0.29 |
| 0.75–<1.25 cups | 257 | 0.94 cup | 24.7 ± 0.40 | 23.3 ± 0.45 | 26.0 ± 0.63 |
| ≥1.25 cups | 120 | 1.5 cups | 24.1 ± 0.58 | 22.2 ± 0.70 | 25.7 ± 0.87 |
| *p*-trend | | | 0.0022 | 0.001 | 0.058 |

* All girls model is adjusted for race. BMI: body mass index; s.e.: standard error.

## 4. Discussion

The results of these analyses suggest that higher intakes of 100% fruit juice during preadolescence were associated with higher intakes of both whole fruit and total fruit as well as better overall diet quality throughout adolescence as measured by total scores on the HEI. Black girls consumed less total fruit at all ages than white girls, and this difference was due to lower intakes of whole fruit. The positive association between fruit juice intake and later diet quality was evident in all girls, regardless of race. Both white and black girls who consumed more 100% fruit juice during preadolescence were also more likely to meet the DGA recommendations for whole fruit intake throughout adolescence. Finally, girls with higher intakes of preadolescent fruit juice had lower BMIs during adolescence and at the end of adolescence than girls who did not drink fruit juice in the preadolescent years. The beneficial effects of whole fruit consumption on BMI were similar.

The intake of total fruit, and particularly whole fruit, has increased in recent years among younger children [8]. However, this is not the case in older children. Previous studies have shown that 14–18-year-olds generally consume only half of the recommended amount of whole fruit per day [4,25]. According to the DGA, the proportion of total fruit consumed that is derived from fruit juice declines with age throughout the life span, with nearly half of total fruit coming from fruit juice among preschoolers, while only one third of total fruit is derived from fruit juice among adults. By late adolescence, total fruit intake is only about half of what is recommended [4], suggesting that the identification of strategies for promoting total fruit consumption is an important priority during the childhood years.

Whole fruit contains many essential vitamins and minerals and is an important source of dietary fiber. Thus, it is an important part of a healthy diet [26]. These analyses support those of other studies showing that fruit juice consumption is associated with a higher diet quality among children and adolescents [10]. While fruit juices provide limited amounts of dietary fiber, they do contain important quantities of magnesium and potassium [13]. Some juices, such as orange juice, have been shown to have much higher levels of bioavailability for both carotenoids and flavonoids than whole fruit [27]. Thus, the beneficial properties of whole fruit and fruit juice, including their antioxidant and anti-inflammatory properties, may differ [28,29]. Data from the DGA show that the most commonly consumed whole fruits are apples, bananas, watermelon, grapes, and strawberries, while the most commonly consumed fruit juice, by far, is orange juice [19]. Thus, whole fruit and fruit juice may have different roles in the prevention of cardiovascular disease, diabetes, and obesity, suggesting perhaps that a balanced intake of whole fruit and fruit juice may provide an optimal nutrient profile [30].

Fruit juice consumption has been at the center of controversy regarding the promotion of excess weight gain in children. In our own previous analyses using data from the Framingham Children's Study, we found no association between the consumption of 100% fruit juice starting in preschool and the change in BMI throughout childhood [31]. A 2017 meta-analysis found that one 6–8-ounce serving per day of 100% fruit juice in children ages 1–6 years led to a 0.087 unit increase in BMI z-scores [32]. However, this same amount of

fruit juice had no impact on BMI in children ages 7–18 years. The results of the current study support those of earlier studies among adolescents, including an analysis from the Growing Up Today Study II, which found that fruit juice consumption among 9–16-year-old girls was inversely associated with a change in BMI [15,16]. In the Women's Health Study, middle-aged and older women who had higher total fruit intakes had a lower risk of overweight and obesity [33]. In these analyses, we also found an inverse association between total fruit intake and BMI at the end of adolescence. This association was very similar to that for 100% fruit juice. Because fruit juice and total fruit intake are highly correlated (Pearson $r = 0.64$ at baseline and 0.72 in late adolescence), it is difficult to separate out the effect of fruit juice from total fruit. However, the weaker correlation between fruit juice and whole fruit intake at baseline ($r = 0.07$) and 0.12 in later adolescence) suggests that whole fruit itself may not be responsible for the observed beneficial association between fruit juice and BMI.

The current study has several important strengths in terms of its design. It is a relatively large prospective study with dietary data derived from multiple sets of three-day diet records, which should yield more precise estimates of dietary intake than most other methods used in large-scale epidemiologic studies. Additionally, the sample size allowed for a comparison of black and white adolescent girls. On the other hand, at the outset of the study, the girls were 9–10 years of age when accurate quantification of the amounts consumed is challenging. While the children were encouraged to obtain details of recipes and food preparation from a parent, there is no guarantee that this was consistently done. All self-reported dietary assessment methods are prone to error. However, given that these data were collected at a time when there was little concern about potential adverse effects of fruit juice consumption, the reported intakes of fruit and fruit juice are unlikely to be biased.

A 2017 commentary concluded that further evidence is needed to refine the recommendations for fruit juice consumption during childhood [34]. In the interim, they concluded that there is no justification for banning fruit juice other than during the first year of life. The current study adds evidence that may provide support for a role of fruit juice in the evolution of healthy eating behaviors without adversely impacting weight gain. These data suggest that fruit juice consumption could promote later intake of whole fruit through its impact on taste preferences because taste perception develops throughout childhood and seems to stabilize in midadolescence [35]. The taste of a variety of fruit juices tends to be acceptable to young children and may, through early exposure, facilitate the development of preferences for a variety of whole fruits. Since fruit juice is more available (regardless of climate and season), has a longer shelf life, and is often more affordable than many whole fruits, it may play a particularly important role in meeting DGA recommendations for families of a lower socioeconomic status who typically have lower intakes of total fruit and whole fruit [7].

The current data provide evidence supporting a beneficial association between early juice-drinking behaviors and the development of healthy dietary behaviors while having no apparent adverse impact on adolescent BMI.

**Supplementary Materials:** The following supporting information can be downloaded at: https:// www.mdpi.com/article/10.3390/beverages9020042/s1, Table S1: Median intakes and interquartile ranges for white and black girls for fruit juice, whole fruit, and total fruit throughout adolescence; Table S2: Mean (±s.e.) intakes of total fruit, whole fruit, and 100% fruit juice at each age according to category of 100% fruit juice intake at 9–10 years of age; Table S3: Mean BMI at 17–20 years of age according to intake of whole fruit and total fruit at 9–10 years of age.

**Author Contributions:** Conceptualization, L.L.M.; methodology, L.L.M. and M.R.S.; formal analysis, M.R.S.; investigation, S.R.D. and L.L.M.; data curation, M.R.S., S.R.D. and L.L.M.; writing—original draft preparation, L.L.M.; writing—review and editing, X.Z., L.W., M.R.S., M.L.B., S.R.D. and L.L.M.; supervision, project administration, and funding acquisition, S.R.D. and L.L.M. All authors have read and agreed to the published version of the manuscript.

**Funding:** This research was funded by National Institute of Diabetes and Digestive and Kidney Diseases (NIDDK), grant number #R21 DK075068. Additional funding from the Juice Products Association.

**Data Availability Statement:** The data used in this study are publicly available through the National Heart, Lung, and Blood Institute's (NHLBI) BioLINCC repository.

**Conflicts of Interest:** The authors declare no conflict of interest.

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
