# Peer review of "Fruit Juice Consumption, Body Mass Index, and Adolescent Diet Quality in a Biracial Cohort"

_beverages, doi:10.3390/beverages9020042_

Round 1

Reviewer 1 Report

The study was  properly conducted.

However, in the abstract, the authors mentioned that 1921 black and white girls, ages 9-10 with years of follow-up for diet and BMI. According to what is in the text,  2165 girls were for analyses of juice intake and BMI.

Reviewer 2 Report

Summary

The present study evaluates girls’ 100% fruit juice intake at 9-10 years and associates it with fruit intake in adolescence and BMI. The research question is interesting and more research on this topic is needed to clarify confusing dietary guidelines for families. However, more detail is needed to evaluate how meaningful the present results are. The manuscript currently lacks a hypothesis(or hypotheses). More detail is needed in the methods especially around what were planned comparisons. The figures lack a description of variance.  

Major

What was/were the hypotheses?

Why was greater than 1.25 cups chosen as the second cut off? I don’t see justification for using both sets of categories.

The methods describe a "first analytic question", but the lack of hypotheses make it difficult to understand how many models were run and why

What correction for multiple comparisons was used?

Figure 1 should have some indication of the variation in the mean or median at each time point (SD or SE)

The total fruit seems much higher than expected in figure given the medians for the whole fruit and 100% fruit juice. Were these from the HEI categories?

Figure 2 additionally needs to indicate what the variance is at each time point

It would be helpful to include a table with the RM ANOVA results and to additionally indicate in figure 2 which time points are significantly different and between which groups. 

Figure 2 indicates that the relationship of time to fruit intake (all types) is non-linear. Why were linear models used?

The covariate analyses needs to be included in supplemental materials

I recommend using a box plot or violin plot instead of the histogram for figure 3. This would improve the reader’s ability to understand the distribution, error, and mean (or median)

The cell sizes between fruit juice groups are fairly different, is the variance within groups similar? Otherwise I am not sure this analysis meets the assumptions for an rmANOVA. Additionally, was the drop out of data similar between groups. How was missing data modeled over time?

Many of the results compare the highest juice intake group with the lowest juice intake group, however none of the contrasts are outlined in the methods. How were the contrasts chosen? What methods were used to correct for multiple comparisons both in-within a model and between models?

It would be interesting to know how the juice intake is related to whole fruit intake. Does whole fruit intake at 9-10 predict total fruit intake? Is the effect of the juice or the whole fruit intake at 9-10 years more important?

In the BMI analysis, since juice was correlated with total and whole fruit, are you seeing the effect in figure 4 from the juice or from total fruit? 

Minor

Need to define DGA in abstract

Figure legends are difficult to read due to background 

Round 2

Reviewer 2 Report

While I appreciate the author's responses to some of my review. I am puzzled by the resistance to add some indication of variation to their plots. Including variance to plots is standard and is increasingly recommended in the field (https://journals.physiology.org/doi/full/10.1152/ajpheart.00071.2020 see table 2). If the concern is how the plots look there are a variety of methods to jitter the points to maintain clarity. 

Additionally in Figure 2, if it was modeled linearly, then why not plot the results as a scatter plot with the fitted line and indication of variance. Using ggplot2 this can be accomplished with the geom_smooth function and you can input the model as the line. This would provide an idea of what the data actually looks like and what the fitted line looks like. Further, the authors mention the trends appearing linear, however this can be tested. The model could have been fitted with a spline or using generalized additive modeling and compared to the linear. Justification for choices should be provided in the methods.

Additionally I find it odd that the authors do not want to include their covariate analyses in the supplemental materials. There are quite a few tables and figures that is true, and that is the purpose of the supplemental materials. 

For figure 3, using a violin plot (or at the very least a box plot) would not require 40 additional plots. The exact same figure can be generated using ggplot2 or matplotlib but with a better representation of the data using a violin plot. At the very least, the data points should be super imposed on the bars to give an idea of the distribution. 

Additionally, the authors note that they used a mixed linear model but this is not described in the methods. I assume they used some kind of random intercept or slope but it isn't clear. Further they note they restricted analyses to those with non-missing data but this does not address my question of drop out between groups. The mixed linear model should be described in the methods. This also loops back to figure 2. If a mixed linear model was used, then figure 2 should be a scatter plot with the fitted lines.

Given that the author's don't prefer to correct for multiple comparisons, this should be mentioned in the statistical analyses. They can add the citations they provided in the response to the authors. 

To clarify the comment the authors did not understand:

The Figure legends are difficult to read due to the background lines in the plot cutting through the text. This can be fixed by filling the background of the legend with a solid (white) background color.
